# Metagenomic Signatures of Colorectal Cancer in the Jordanian Population: A Regional Case-Control Analysis Using 16S rRNA Profiling

**DOI:** 10.3390/microorganisms13081963

**Published:** 2025-08-21

**Authors:** Lo’ai Alanagreh, Minas A. Mustafa, Mohammad Borhan Al-Zghoul, Muhannad I. Massadeh, Osamah Batiha, Maher Sughayer, Rashed Taiseer Almashakbeh, Haya Bader Abu Suilike, Faten S. Tout, Foad Alzoughool

**Affiliations:** 1Department of Medical Laboratory Sciences, Faculty of Applied Medical Sciences, The Hashemite University, Zarqa 13133, Jordan; minasahmad36@gmail.com (M.A.M.); hayabaderabusuilike@gmail.com (H.B.A.S.); fatenj@hu.edu.jo (F.S.T.); foad@hu.edu.jo (F.A.); 2Department of Medical Laboratory, Faculty of Allied Medical Sciences, Zarqa University, Zarqa 13110, Jordan; 3Department of Basic Medical Veterinary Sciences, Faculty of Veterinary Medicine, Jordan University of Science and Technology, Irbid 22110, Jordan; alzghoul@just.edu.jo; 4Department of Biology and Biotechnology, Faculty of Science, The Hashemite University, Zarqa 13133, Jordan; massadeh@hu.edu.jo; 5Department of Biotechnology and Genetic Engineering, Jordan University of Science and Technology, Irbid 22110, Jordan; oybatiha@just.edu.jo; 6King Hussein Cancer Center, 202 Queen Rania Al-Abdullah Street, Al-Jubaiha, Amman 11941, Jordan; msughayer@khcc.jo (M.S.); ra.14996@khcc.jo (R.T.A.); 7Department of Nursing, Faculty of Health Sciences, Higher Colleges of Technology, Abu Dhabi P.O. Box 25026, United Arab Emirates

**Keywords:** colorectal cancer, microbiome, metagenomic analysis, dysbiosis, Jordan, next-generation sequencing

## Abstract

The gut microbiota plays a pivotal role in developing colorectal cancer (CRC) through interactions with host immunity, metabolism, and inflammation. However, microbiome-based studies remain scarce in Middle Eastern populations, limiting regional insights into microbial signatures associated with CRC. This study aimed to characterize the gut microbiota profiles of Jordanian CRC patients using 16S rRNA gene sequencing and compare them to those of healthy controls from the GutFeeling KnowledgeBase (GutFeelingKB). Stool samples from 50 CRC patients were analyzed using Illumina iSeq targeting the V3–V4 region. Taxonomic profiling was conducted with a standardized 16S metagenomics pipeline and compared with GutFeelingKB reference data. CRC samples were enriched in *Streptococcus*, *Enterococcus*, *Klebsiella*, *Escherichia*, *Citrobacter*, *Veillonella*, *Megamonas*, and *Eggerthella*, while beneficial butyrate-producing genera such as *Roseburia, Ruminococcus*, *Akkermansia*, *Faecalibacterium*, and *Bacteroides* were significantly depleted. The absence of *Fusobacterium nucleatum* and *Bacteroides fragilis*—commonly seen in global studies—suggests region-specific microbial patterns. This study is the first metagenomic study profiling CRC-associated microbiota in Jordan. The findings reveal a dysbiotic microbial signature that reflects both global changes associated with CRC and local ecological influences. This research emphasizes the importance of population-specific microbiome studies and highlights the need to include appropriately matched controls in future investigations.

## 1. Introduction

The human gastrointestinal (GI) tract is inhabited by an extensive and intricate ecosystem of microorganisms, commonly known as the gut microbiota [1]. This community consists of trillions of microbes, including bacteria, archaea, viruses, and fungi, with bacteria being the most extensively studied due to their significant roles in human health and disease [2]. In healthy individuals, these microorganisms perform a variety of essential physiological functions, such as aiding in the digestion of complex carbohydrates, producing short-chain fatty acids (SCFAs), synthesizing vitamins, and modulating the host’s immune system [3]. The dominant bacterial phyla found in the gut are Firmicutes and Bacteroidetes. Other significant groups include Actinobacteria, Proteobacteria, and Verrucomicrobia. The relative abundance of these phyla is often assessed through the Firmicutes/Bacteroidetes (F/B) ratio, which has been suggested as a potential marker of gut health and may reflect the overall balance of microbial populations [4]. However, it is essential to recognize that the composition of the gut microbiota is highly individualized and influenced by a multitude of factors. These include host genetics, age, geographical location, dietary habits, exposure to antibiotics, and various environmental influences [5]. Such variability underscores the complexity of the gut microbiome and its significant role in human health, suggesting that personalized approaches may be necessary to understand and promote gut health in different populations.

Disruptions to the normal composition of the gut microbiota, a condition referred to as dysbiosis, have been linked to a variety of diseases, including inflammatory bowel disease, obesity, diabetes, autoimmune disorders, and notably, CRC. In the context of CRC, dysbiosis may promote the development of cancer through several interconnected mechanisms, highlighting the complex interaction between microbial activity and the host’s responses [6,7].

Chronic inflammation is a key pathway associated with cancer development. Certain microbial taxa are known to induce persistent low-grade inflammation within the intestinal epithelium. For Example, *Fusobacterium nucleatum* is known to activate pro-inflammatory pathways via Toll-like receptor 4 (TLR4), which can foster an environment conducive to tumor growth [8,9]. Additionally, this microorganism also recruits myeloid-derived suppressor cells and other immunosuppressive elements, further facilitating tumor progression [9]. Another significant mechanism is the production of genotoxins. For example, strains of *Escherichia coli* possessing the pks genomic island produce colibactin, a genotoxin that induces DNA double-strand breaks in host cells. This DNA damage can result in mutations and chromosomal instability, ultimately initiating and promoting colorectal tumorigenesis [10]. Moreover, metabolic alterations and epigenetic modifications play vital roles in this context. Microbial metabolites, particularly short-chain fatty acids (SCFAs) such as butyrate, are utilized as energy sources for colonocytes and inhibit histone deacetylases (HDACs), which promotes the expression of anti-inflammatory genes and tumor-suppressive pathways. A depletion in butyrate-producing bacteria (e.g., *Roseburia*, *Faecalibacterium*, *Eubacterium*, *Ruminococcus*) has been linked to CRC [11,12]. This correlation has consistently been observed in CRC patients, emphasizing the protective role of a SCFA-rich microbial community [13]. Finally, immune dysregulation driven by microbiota alterations can impact mucosal immune tolerance and surveillance [14]. The loss of key commensals or the expansion of pathobionts may skew immune responses towards a pro-tumorigenic state [15]. Changes in the microbiota might impair T-cell mediated tumor immunosurveillance and foster regulatory or exhausted immune cell phenotypes, which enable tumor cells to evade immune destruction [16].

An increasing number of studies involving both human and animal subjects support the notion that microbial signatures may play either a causal or an accompanying role in the development of CRC. For example, a meta-analysis by Wirbel et al. (2019) identified a consistent set of microbial markers present in CRC patients across diverse geographic regions [17]. This analysis highlighted an enrichment of bacteria such as Fusobacterium, Peptostreptococcus, and Parvimonas, alongside a decrease in species that produce short-chain fatty acids (SCFAs). Subsequent large-scale investigations have further expanded the catalog of CRC-associated microbes, revealing many previously unrecognized bacterial candidates linked to the disease [18]. However, these patterns are not uniform across all populations. Studies in Chinese, Japanese, and European cohorts show region-specific microbial imbalances, likely due to variations in dietary fiber intake, meat consumption, antibiotic use, and other lifestyle factors [19,20]. This variability highlights the importance of contextual microbiome research, especially in underrepresented regions like the Middle East. Jordan represents a unique and underrepresented population for CRC microbiome research due to its distinct dietary patterns, environmental exposures, and genetic background. Traditional Jordanian diets, characterized by high consumption of olive oil, whole grains, legumes, and fermented dairy products, differ markedly from Western dietary profiles and may shape a unique gut microbial landscape. Additionally, regional variations in antibiotic use, water sources, and environmental exposures may further impact gut microbiota composition. The Arab population, including Jordanians, also exhibits specific genetic polymorphisms that may influence host–microbiota interactions and CRC susceptibility. Despite rising CRC incidence in the region, there is a significant gap in microbiome data from Middle Eastern populations. By studying the Jordanian cohort, this research aims to address this gap and contribute region-specific insights to the global understanding of CRC-associated microbiota.

Furthermore, the application of 16S rRNA amplicon sequencing, commonly used in microbiome studies, enables cost-effective taxonomic profiling at approximately the genus level. While this technique is efficient for surveying community composition, it lacks the resolution to identify strain-level variation or to characterize functional genes involved in CRC pathogenesis. Advanced methods such as shotgun metagenomics and metatranscriptomics are increasingly preferred for gaining mechanistic insights [21], although they remain less accessible in low-resource settings. Recent comparisons indicate that while both 16S and shotgun approaches can capture major community shifts, shotgun sequencing provides greater taxonomic depth and functional insight into the microbiome [22].

In summary, the gut microbiota is intricately connected to colorectal health and disease. Understanding its structure and function in diverse populations can help uncover microbial biomarkers, inform diagnostic strategies, and even pave the way for microbiota-based therapeutics. Yet, regional data—especially from Arab populations—remain scarce. Addressing this gap is the central motivation of the present study.

## 2. Materials and Methods

### 2.1. Study Design and Ethical Approval

This study was designed as a comparative cross-sectional analysis of gut microbiota in CRC patients versus healthy controls. Ethical approval for this research was granted by the Institutional Review Board (IRB) at Hashemite University, under protocol number 19/7/2022/2023, dated 6 September 2023. The CRC samples were anonymized and derived from previously diag-nosed cases, thus not requiring individual informed consent. All procedures adhered to the ethical principles outlined in the Declaration of Helsinki.

A major limitation of this study is the absence of a locally recruited, demographically matched healthy control group. Instead, we utilized healthy reference profiles from the GutFeelingKB database, which may differ from the Jordanian population in terms of region, diet, lifestyle, and sequencing protocols. These differences may introduce population and technical biases, potentially confounding observed microbiome differences attributed to CRC. Future studies should prioritize the recruitment of local controls to enable more accurate, population-specific comparisons.

### 2.2. Sample Collection

A total of 50 stool samples were collected from CRC patients diagnosed and treated at the King Hussein Cancer Center (KHCC) in Jordan. Samples were obtained using sterile, DNA-free containers and immediately transported on ice to the molecular diagnostics laboratory. Upon arrival, specimens were stored at −80 °C to preserve DNA integrity until further processing. Clinical metadata such as patient age, sex, and cancer stage were not linked to individual samples due to anonymization protocols.

Due to anonymization protocols, individual clinical data (including cancer stage and treatment history) were not available for correlation analyses with microbiome rofiles. Future investigations should strive to collect and integrate clinical metadata to better elucidate relationships between microbial alterations and patient characteristics.

### 2.3. DNA Extraction and Quantification

Bacterial DNA was extracted from approximately 250 mg of each stool sample using the QIAamp^®^ PowerFecal^®^ Pro DNA Kit (Qiagen, Hilden, Germany), following the man-ufacturer’s protocol. The extraction procedure included a mechanical lysis step with bead-beating to ensure adequate disruption of Gram-positive bacterial cell walls and maximize DNA yield. DNA was eluted in 10 mM Tris-HCl (pH 8.5) and quantified using the Qubit™ dsDNA High Sensitivity Assay Kit (Thermo Fisher Scientific, Waltham, MA, USA). Purity of DNA was confirmed by measuring absorbance ratios (260/280 nm) on a NanoDrop spec-trophotometer, with all samples showing ratios ~1.8–2.0 indicating high-quality DNA.

### 2.4. 16S rRNA Amplicon Library Preparation

Library preparation targeted the V3–V4 hypervariable regions of the bacterial 16S rRNA gene using the following primer set. Forward primer (341F) [5′-CCTACGGGNGGCWGCAG-3′], and reverse primer (805R) [5′-GACTACHVGGGTATCTAATCC-3′]. PCR amplification was performed using a KAPA HiFi HotStart ReadyMix in a thermal cycler under the following conditions: initial dena-turation at 95 °C for 3 min; 25 cycles of 95 °C for 30 s, 55 °C for 30 s, 72 °C for 30 s; and a final extension at 72 °C for 5 min. Amplicons (~550 bp) were purified using AMPure XP beads (Beckman Coulter, Brea, CA, USA) to remove primers and small fragments. This step followed by a second PCR conducted to attach dual indices and Illumina sequencing adapters (Nextera XT Index Kit, Illumina, San Diego, CA, USA). After indexing, a second round of AMPure XP purification was performed to eliminate any remaining primer-dimers or free adapters.

### 2.5. Library Quantification and Pooling

Indexed libraries were quantified fluorometrically using a Qubit and normalized to 4 nM. The average library size (~630 bp, including adapters) was verified on an Agilent Bi-oanalyzer 2100 using a High Sensitivity DNA chip. Equimolar amounts of each library were then pooled into a single tube. The pooled library was spiked with 5% PhiX control DNA (Illumina) to increase base complexity and serve as an internal sequencing control.

### 2.6. Sequencing on Illumina iSeq 100 Platform

The pooled library was sequenced on the Illumina iSeq 100 platform (Illumina, San Diego, CA, USA) using a 2 × 150 bp paired-end chemistry run. The iSeq 100 employs Illu-mina’s Sequencing-by-Synthesis (SBS) technology, which enables high-fidelity base in-corporation and real-time detection of fluorescently labeled nucleotides. The run was set up according to the manufacturer’s protocol, including cluster generation and sequencing on the iSeq flow cell. Run quality was monitored with the integrated Illumina BaseSpace Sequence Hub, and predefined quality control (QC) criteria (e.g., >80% of bases with Q30 quality or higher) were met by the completion of the run.

### 2.7. Bioinformatics and Taxonomic Analysis

Raw sequencing reads were processed using Illumina’s 16S Metagenomics workflow, which use the Classify Reads algorithm, an advanced implementation of the RDP Classifier. Instead of the original 8-base words, this approach utilizes 32-base subsequences, enhancing the specificity of species identification. The method employs a naïve Bayesian framework to calculate the conditional probability of a sequence belonging to a specific taxon, enabling classification at both genus and species levels. After demultiplex-ing, reads were quality-filtered (removing reads with ambiguous bases or Phred score < Q20) and merged into contigs. The classifier assigned taxonomy at multiple ranks, using an 80% confidence threshold for classification. QIIME 2 (version 2024.8) was used to per-form additional microbiome analysis to validate and improve the initial taxo-nomic classifications produced by the RDP classifier. Quality control, denoising, and am-plicon sequence variant (ASV) generation were performed using the DADA2 plugin. The ASVs were utilized to generate representative sequences (repseqs), and a robust taxonomic resolution was achieved by taxonomic assignment using the SILVA 138 reference database with 99% similarity criteria. Additionally, Gut-FeelingKB was used to establish a reference baseline for the composition of healthy gut microbes. GutFeelingKB is a curated database of bacterial species frequently observed In the human digestive tract that was created by George Washington University’s Hive Lab [7]. It serves as a valuable resource for metagenomic investigations by encompassing a wide range of microbial taxa typically present in healthy individuals. A heatmap was then generated in R 4.5.1 using the gplots package to visualize microbial species abundance and apply hierarchical clustering across both samples and taxa. To assess within-sample microbial diversity, alpha diversity indices, such as Shannon and Simpson, were computed in R using the vegan package. To show the differences between microbial communities according to Bray–Curtis distances, Principal Coordinates Analysis (PCoA) was carried out in R. The vegan package was used to do the analysis, and the ggplot2 program was used to create the PCoA plot that resulted. To ascertain which microbial genera are most effective at differentiating between the healthy and CRC groups, Linear Discriminant Analysis (LDA) was carried out in R using the MASS package. Using the Wilcoxon test, genera with an LDA score > 2 and a *p*-value < 0.05 were determined to be substantially different between the two groups. Statistical analysis was performed using SPSS v27. Group comparisons of genera abundance were conducted using the Mann–Whitney U test, and results are reported as mean ± standard deviation (SD).

## 3. Results

### 3.1. Overview of Sequencing and Taxonomic Classification

The 16S rRNA sequencing run generated high-quality paired-end reads for 42 CRC patient samples from the original dataset, three samples were lost during sequencing, and five others had too few reads, so they were removed. This left 42 samples for further analysis. After demultiplexing and quality filtering, each sample retained an average of ~45,000 reads suitable for downstream taxonomic analysis. The classification pipeline successfully assigned taxonomy down to the genus level for the majority of sequences (species-level resolution was achieved for some taxa when the V3–V4 region provided sufficient differentiation). As expected, the genus-level assignments were the most reliable, consistent with the known resolution limits of the V3–V4 16S region.

### 3.2. Microbial Community Structure in Healthy Controls and CRC Patients

The phylum-level taxonomic analysis revealed significant variations in microbial composition between the healthy and CRC populations (Figure 1). These alterations extended to the genus level (Figure 2), further supporting the presence of gut microbial dysbiosis associated with colorectal cancer. Notably Firmicutes emerged as the dominant phylum in CRC patients, accounting for approximately 64% of classified reads compared to only 22% in the healthy GutFeelingKB cohort (*p* < 0.05). Within the Firmicutes, the *Streptococcus* and *Enterococcus* genera were markedly enriched in CRC patients. For instance, *Enterococcus* constituted about 7.55% of sequences in CRC samples versus 0.05% in healthy controls, and *Streptococcus* comprised 22.8% in CRC vs. 0.23% in controls (both differences statistically significant, *p* < 0.05, while *Veillonella* and *Megamonas* showed a slight but significant increase in abundance compared to healthy controls (*p* < 0.05). In contrast to the abundance of opportunistic taxa among Firmicutes, beneficial genera such as *Roseburia* were completely missing in CRC samples (0%), as compared to 1.83% in healthy controls. Similarly, *Ruminococcus* and abundance was significantly reduced, accounting for merely 1.45% in CRC patients compared to 4.00% in the control group, while *Faecalibacterium* significant decrease in abundance of CRC compared to healthy controls (*p* < 0.05). Members of the phylum Proteobacteria were also elevated in CRC patients relative to controls. Notably, two genera from Proteobacteria showed increased relative abundance in CRC stools: *Klebsiella* (4.17% in CRC vs. 0.03% in controls) and the *Escherichia/Shigella* group (3.33% in CRC vs. 1.96% in controls). However, good genera from Proteobacteria such *Akkermansia* was significantly reduced. Similarly, *Eggerthella* exhibited a modest yet statistically significant increase in CRC samples relative to healthy individuals (*p* < 0.05). Moreover, *Bacteroides* showed a significant decrease from 65.57% in the healthy controls to only 4.38% in those with CRC. This decline was paralleled by a decrease in *Parabacteroides* and *Bacteroides*. Conversely, *Prevotella* was associated with a significant increase in CRC patients compared to healthy control group (Figure 3).

### 3.3. Microbial Diversity Metrics Reveal Dysbiosis in Colorectal Cancer

Alpha and beta diversity analyses were utilized to assess the structure and complexity of gut microbial communities in CRC patients vs. healthy people. Alpha diversity, as assessed by the Shannon and Simpson indices, was significantly lower in the CRC group, indicating a reduction in microbial richness and evenness. This decrease in microbial diversity is typically linked to an unhealthy gut environment (see Figure 4a). Regarding beta diversity, principal coordinates analysis (PCoA) using Bray–Curtis dissimilarity demonstrated a clear distinction between the CRC and control groups. This difference highlights significant variations in the overall composition of microbial communities, emphasizing the presence of microbial dysbiosis associated with CRC. These changes in microbial diversity and structure indicate that gut microbiota changes may play a role in colorectal cancer development (Figure 4b).

### 3.4. Potential Bacterial Biomarkers Identified via LDA

We used LDA to compare unique microbiomes of CRC patients and healthy individuals. The two groups’ bacterial compositions differed significantly, as indicated by the LDA values (Figure 5). Beneficial commensals, such as *Akkermansia*, *Faecalibacterium*, *Bacteroides*, *Parabacteroides* and *Ruminococcus*, which are frequently linked to the integrity of the gut barrier, the synthesis of short-chain fatty acids, and anti-inflammatory qualities, predominated in the microbiome of the healthy group. Conversely, the CRC group showed an enrichment of taxa that may be harmful or pro-inflammatory, including *Megamonas*, *Eggerthella*, *Citrobacter*, *Enterococcus*, *Streptococcus*, *Klebsiella*, *Prevotella* and *Veillonella*. These results support the possible involvement of the gut microbiota in disease etiology and draw attention to the unique microbial profiles linked to colorectal cancer.

## 4. Discussion

Jordan represents a unique and underrepresented population for CRC microbiome research due to its distinct dietary patterns, environmental exposures, and genetic background. Traditional Jordanian diets, characterized by high consumption of olive oil, whole grains, legumes, and fermented dairy products, differ markedly from Western dietary profiles and may shape a unique gut microbial landscape. Additionally, regional variations in antibiotic use, water sources, and environmental exposures may further impact gut microbiota composition. The Arab population, including Jordanians, also exhibits specific genetic polymorphisms that may influence host–microbiota interactions and CRC susceptibility. Despite rising CRC incidence in the region, there is a significant gap in microbiome data from Middle Eastern populations. By studying the Jordanian cohort, this research aims to address this gap and contribute region-specific insights to the global understanding of CRC-associated microbiota.

This study, which includes microbial profiling of the gut microbiota and a comparison of CRC patients in the Jordanian population to healthy individuals from the GutFeelingKB database, highlighting both overlaps and contrasts with global microbiome literature. The gut microbiota of CRC patients had a higher abundance of the Firmicutes phylum compared to healthy controls. These results are consistent with [7] A variety of genera in the phylum Firmicutes are known to have an impact in preserving intestinal homeostasis. The relative abundance of the species *Enterococcus* was significantly higher in CRC patients than in healthy controls, according to the current study (*p* < 0.05). This result is consistent with earlier studies that found Enterococcus to be more common in stool samples from people with colorectal cancer than from healthy people [7,23]. Likewise, our research showed that CRC patients had a significantly higher genus *Streptococcus* enrichment than healthy people (*p* < 0.05). This is in line with findings of Mira-Pascual et and Zhang et al., who similarly noted noticeably greater *Streptococcus* levels in CRC populations [24,25]. The elevated presence of *Streptococcus* and *Enterococcus*, aligns with prior reports indicating their pro-inflammatory and mucin-degrading capabilities in the colorectal mucosa [7,26]. The *Streptococcus* genus exhibits both pro- and anti-tumorigenic roles in colorectal cancer. Members of the *Streptococcus bovis*/*Streptococcus equinus* complex—particularly *S. gallolyticus*—are strongly associated with CRC, with studies showing a significantly increased risk in patients with SBSEC bacteremia [27,28]. *Enterococcus faecalis* contributes to colorectal cancer by producing reactive oxygen species that damage DNA and promote genomic instability. It also secretes metalloproteases that weaken the intestinal barrier and trigger inflammation, fostering a pro-tumorigenic environment [29]. Additionally, *Veillonella* and *Megamonas*—genera previously associated with colorectal tumorigenesis and inflammatory bowel disease, further supporting their involvement in CRC-related gut dysbiosis [30,31].

*Bacteroidetes* emerged as the most abundant bacterial phylum in stool samples from CRC patients compared to healthy individuals [32]. However, our findings revealed a relative decrease in Bacteroidetes in patients with CRC. Interestingly, a high-fat diet (HFD) may alter gut microbial composition by boosting Firmicutes while decreasing Bacteroidetes [33]. The decreased abundance of Bacteroidetes in our CRC sample could indicate microbial dysbiosis caused by dietary and lifestyle factors unique to the Jordanian population. Interestingly, the most notable decline within the Bacteroidetes phylum was observed in the *Bacteroides* genus. In contrast, *Prevotella* exhibited a significant increase in abundance among CRC patients compared to healthy controls (*p* < 0.05). Previous studies have demonstrated that *Prevotella* can activate Th17-mediated immune responses, which are closely associated with mucosal inflammation [34].

The enrichment of Proteobacteria (notably *Klebsiella* and *Escherichia*) further supports a disease-associated microbial shift. Both genera include strains that can produce pro-inflammatory lipopolysaccharides (LPS) and genotoxins such as colibactin, leading to DNA damage and chronic inflammation—two recognized hallmarks of cancer [10,35]. *Klebsiella* species are opportunistic pathogens that may exploit the tumor-altered gut environment to proliferate. Their presence in high numbers could exacerbate inflammation through endotoxin release [36]. Similarly, *E. coli* carrying the pks island can directly contribute to mutagenesis in colonic cells [37,38]. Moreover, Citrobacter was shown to be abundant in CRC patients. This organism has been demonstrated to cause inflammation via Th1/Th17 immunological pathways, fostering a pro-inflammatory environment favorable to tumor formation [39]. The elevated levels of these Proteobacteria in Jordanian CRC patients mirror findings from other populations in terms of the functional implications, even if the exact species or strains might differ.

Furthermore, *Eggerthella*, a genus within the Actinobacteria phylum and recognized as a microbial marker for colorectal cancer, was notably enriched in CRC samples, suggesting its potential involvement in tumor progression [40,41].

Butyrate plays a central role in maintaining epithelial homeostasis, inducing apoptosis of malignant cells, and modulating immune responses [42]. The marked reduction in butyrate-producing genera, including *faecalibacterium*, *Roseburia* and *Ruminococcus*, suggests a weakened gut barrier and reduced anti-inflammatory metabolic activity [11,43]. A low-butyrate environment in the colon can lead to impaired energy supply to colonocytes and may favor a microenvironment of uncontrolled inflammation and cell proliferation. The loss of these protective commensals in our CRC cohort is in line with studies worldwide that document reduced SCFA levels in the stool of CRC patients [13,42]. This deprivation may permit oncogenic bacteria to colonize and persist, essentially removing a layer of colon cancer defense [44]. *Akkermansia* has demonstrated promising anticancer effects in CRC. It boosts anti-tumor immune responses, inhibits tumor cell proliferation, promotes apoptosis, and prevents tumor cell escape [45]. Notably, a decrease in the abundance of A. muciniphila has been associated with various systemic illnesses, including obesity, diabetes, and intestinal inflammation [46].

Interestingly, *Fusobacterium nucleatum*, commonly found in CRC cohorts from the United States, Europe, and East Asia [8,17], was not detected in this Jordanian dataset. *F. nucleatum* is a prominent CRC-associated bacterium known for aggregating within tumors and modulating the tumor immune microenvironment [47]. The absence of *Fusobacterium nucleatum* in our CRC samples should be interpreted with caution, as the comparison group was sourced from GutFeelingKB, a Western population. Without analysis of healthy Jordanian controls, it is unclear whether the absence reflects regional baseline differences, methodological limitations, or a true deviation from global CRC patterns. Future research must prioritize recruitment of local, matched healthy controls to resolve this uncertainty. It is possible that in Middle Eastern populations, other bacteria fulfill the role that *Fusobacterium* plays in Western cohorts. Notably, the microbes that were abundant in our patients (like *Streptococcus* and *Klebsiella*) could be the key drivers of dysbiosis in this population, rather than *Fusobacterium*. This population-specific difference underscores the importance of conducting local microbiome studies; global “one-size-fits-all” microbial markers may not be universally applicable [18].

This study underscores the critical need for microbiome profiling in underrepresented populations like the Arab world, where unique regional factors profoundly influence microbial ecosystems. Traditional Jordanian diets, rich in olive oil, whole grains, and legumes, differ significantly from Western diets and play a pivotal role in shaping gut microbiota composition—high-fiber foods promote SCFA-producing bacteria, while high-fat diets favor pro-inflammatory microbes [48]. Additionally, variations in early-life factors such as childbirth practices, breastfeeding, and antibiotic use can establish distinct baseline microbiomes with long-term implications for gut health and immunity [49]. Moreover, genetic polymorphisms common in Arab populations may affect how the microbiota interacts with the host, potentially increasing susceptibility to specific pathogens or inflammation-driven conditions [50]. The dominance of Klebsiella and Streptococcus in our CRC patients may reflect these regional influences. It is plausible that local environmental exposures (e.g., prevalent microbes in water or food) and lifestyles (e.g., communal eating practices, use of traditional fermented foods) select for a gut microbiome that, when perturbed by cancer, shows overgrowth of these particular genera. Identifying these patterns is crucial: if Klebsiella and Streptococcus are important in Jordanian CRC, interventions (dietary modifications, probiotics) could be tailored to target these organisms. Our work adds a piece to the emerging puzzle of how geography and lifestyle intersect with the gut microbiome in cancer.

In this study, we utilized the GutFeelingKB database as a source of microbiome data from healthy control individuals [51]. While GutFeelingKB provides a comprehensive catalog of gut microbiota profiles from healthy populations, its use as a control group introduces interpretive challenges, particularly because regional, dietary, and demographic variations can significantly influence the gut microbiome independent of disease status. In the absence of local healthy controls, it becomes challenging to ascertain whether the observed microbial alterations stem from colorectal cancer (CRC) pathology or pre-existing differences between the Jordanian and Western populations represented within GutFeelingKB.

Furthermore, the lack of exhaustive metadata from the GutFeelingKB cohort restricts the capacity to conduct sensitivity analyses or appropriately match controls. Such limitations necessitate caution in attributing observed differences solely to CRC. Notwithstanding these constraints, employing GutFeelingKB represents a pragmatic approach given the unavailability of locally collected Jordanian healthy samples. The pronounced distinctions observed between CRC and GutFeelingKB profiles, such as significant disparities in Bacteroides abundance, likely surpass what technical variation could account for, indicating genuine disease-related shifts. In light of these factors, it is essential for future research endeavors to prioritize the recruitment of healthy Jordanian participants, ideally matched for parameters such as age, gender, and dietary habits, to serve as controls. This approach will enhance the validity of microbiome comparisons and facilitate the distinction between alterations that are truly indicative of disease and those that can be attributed to regional baseline differences. A key limitation of 16S rRNA V3–V4 amplicon sequencing is its insufficient resolution for species- and strain-level identification. This precludes differentiation between pathogenic and non-pathogenic strains, such as Escherichia coli carrying the pks island, which are relevant to CRC pathogenesis. Future studies employing shotgun metagenomics or targeted PCR/qPCR assays could provide more granular taxonomic and functional insights.

In this study, functional interpretations regarding microbial metabolites (e.g., SCFAs, genotoxins) are speculative and based solely on the known metabolic capabilities of identified taxa, not on direct functional measurements. Future research should incorporate predictive metagenomics (such as PICRUSt2) or shotgun metagenomic sequencing to more accurately characterize functional alterations in CRC-associated microbiota.

It is important to note that fecal samples were obtained from patients who had already been diagnosed and potentially treated for CRC. Treatments such as chemotherapy, antibiotics, and dietary modifications can significantly alter the gut microbiome. Without detailed clinical metadata, it is not possible to distinguish microbiome changes caused by CRC itself from those induced by treatment or lifestyle adjustments. Future studies should collect and integrate these clinical details to disentangle disease-related microbiome shifts from treatment effects.

Another limitation is that this study relied exclusively on fecal samples. While stool samples provide an accessible snapshot of the gut microbiome, they may underrepresent or miss tumor-adherent bacteria that preferentially colonize the mucosal surface, such as Fusobacterium nucleatum. The absence of such taxa in our results could be due, at least in part, to our sampling strategy. Future studies should incorporate both fecal and tissue-associated samples to obtain a more comprehensive view of CRC-associated microbiota.

Another key limitation is the lack of patient-level metadata, such as age, sex, cancer stage, and comorbidities, due to anonymization protocols. This prevented stratified analyses that could have revealed more nuanced associations between host factors, clinical characteristics, and microbiota profiles. Future studies should integrate detailed metadata to facilitate subgroup analyses and strengthen clinical interpretation.

## 5. Conclusions

This study presents the first metagenomic characterization of the gut microbiota in CRC patients from Jordan, revealing a distinct microbial signature marked by enrichment of pro-inflammatory and opportunistic genera (e.g., *Streptococcus*, *Klebsiella*, *Enterococcu*, *Megamonas*, *Eggerthella*, *Citrobacter*, *Prevotella,* and *Veillonella*.) and depletion of beneficial, SCFA-producing commensals (e.g., *Roseburia*, *Ruminococcus*, *Bacteroides*, *Akkermansia*, *Faecalibacterium*). These findings are broadly consistent with global patterns of CRC-associated dysbiosis [13,17,52] while also reflecting region-specific microbial shifts that may be shaped by dietary, environmental, and genetic factors unique to the Jordanian population.

The observed microbiota alterations suggest a disrupted gut ecosystem conducive to inflammation, immune dysregulation, and carcinogenesis—mechanisms widely implicated in CRC pathophysiology. While the use of the GutFeelingKB database as a control introduced certain limitations, our study still provides compelling evidence that population-specific microbial profiling can yield novel insights and potential biomarkers for CRC. By contributing data from an Arab population, we move toward a more inclusive understanding of the CRC microbiome, which is crucial for translating microbiome research into effective screening tools and therapies worldwide.

## Figures and Tables

**Figure 1 microorganisms-13-01963-f001:**
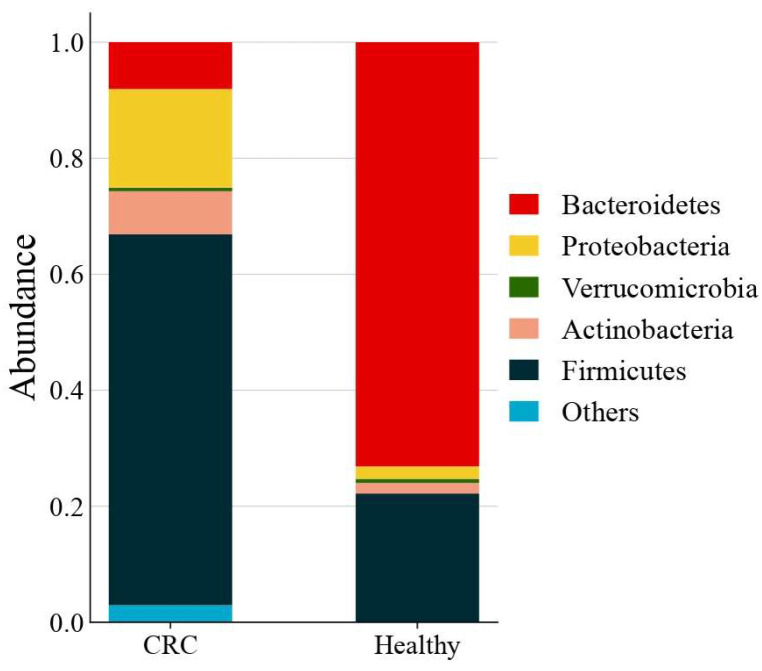
Average gut microbiota composition at the phylum level across all CRC (n = 42) and healthy control samples. Values represent the mean relative abundance for each group.

**Figure 2 microorganisms-13-01963-f002:**
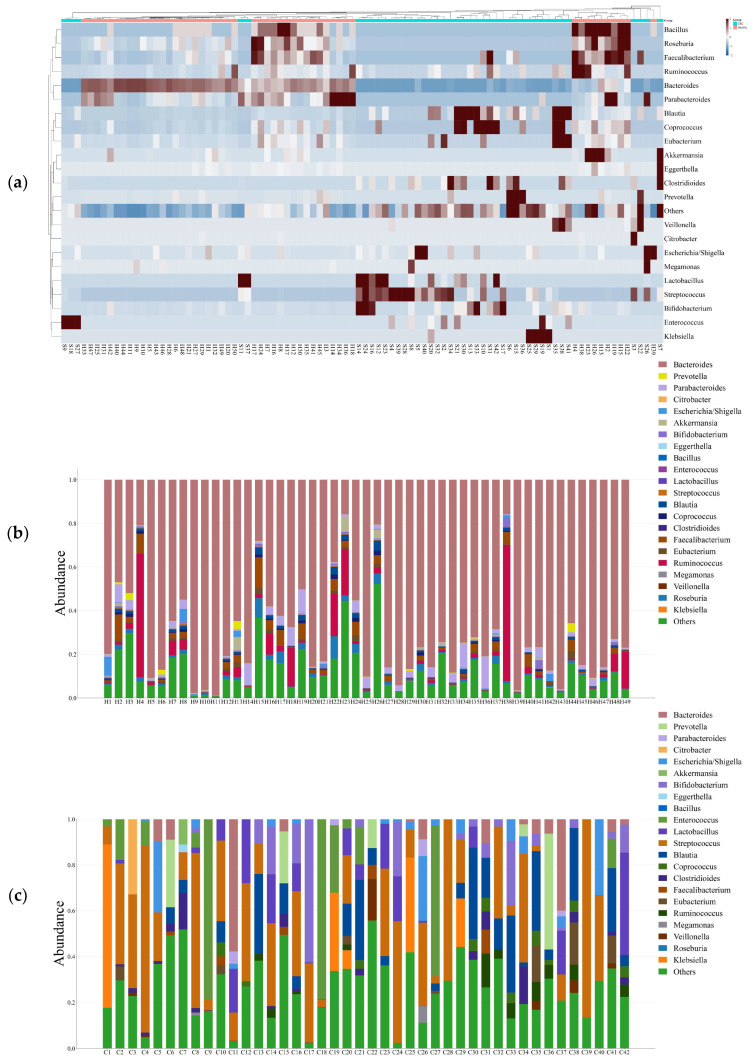
(**a**) Heat map illustrates the differential abundance of Genera between CRC and Healthy groups; (**b**) Bar plot illustrates the abundance of Genera in Healthy groups; (**c**) Bar plot illustrates the abundance of Genera in CRC groups.

**Figure 3 microorganisms-13-01963-f003:**
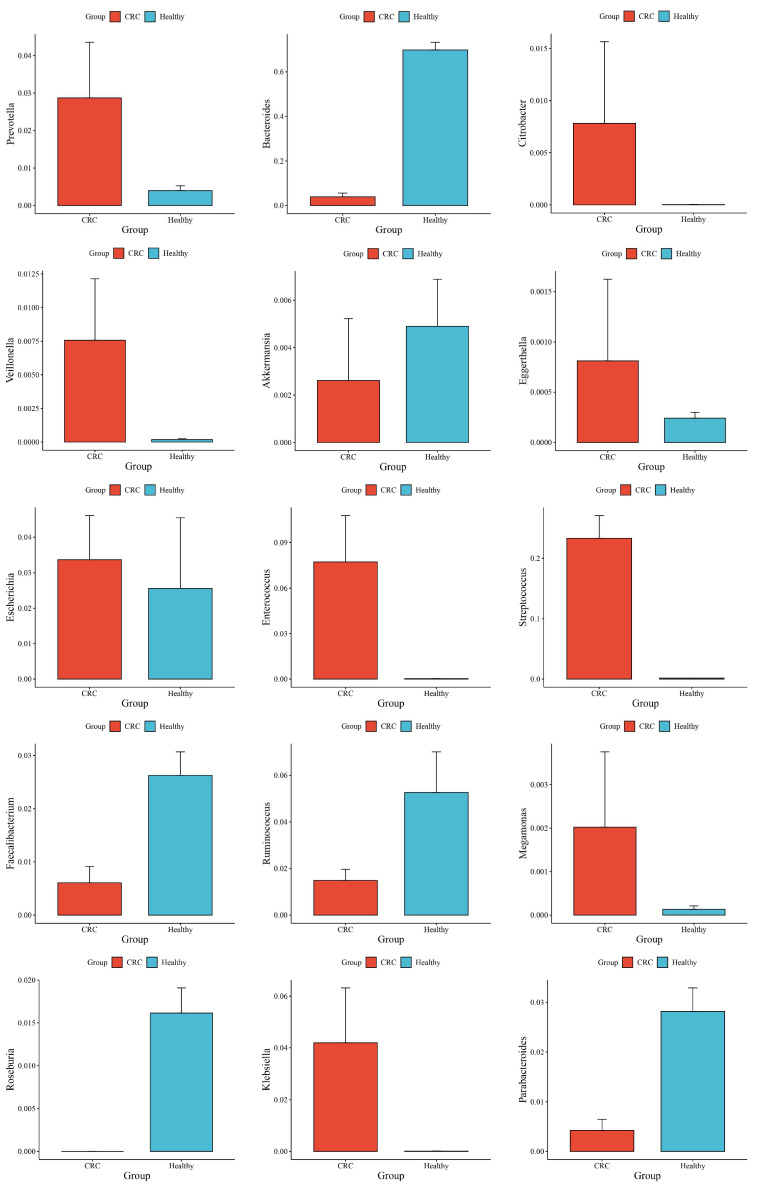
Differences in Mean Abundance and Standard Deviation between CRC and Healthy Groups.

**Figure 4 microorganisms-13-01963-f004:**
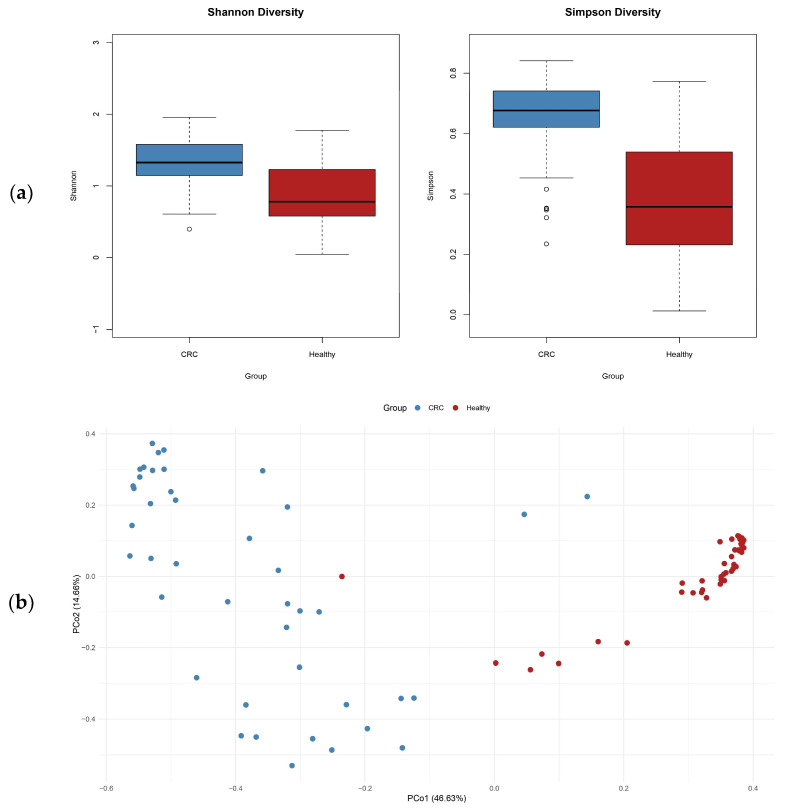
(**a**) Alpha diversity (Shannon and Simpson indices) indicates reduced diversity in CRC; (**b**) PCoA based on Bray–Curtis dissimilarity shows clear separation between groups, highlighting distinct microbial community structures.

**Figure 5 microorganisms-13-01963-f005:**
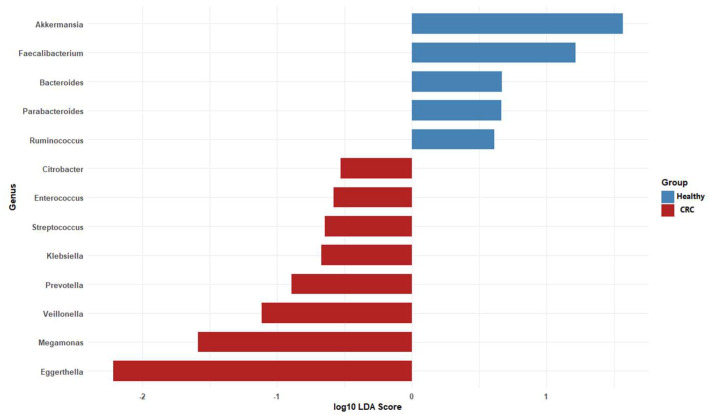
Taxa with both statistical significance (*p* < 0.05, Wilcoxon test) and an LDA score greater than 2 were identified as discriminative features differentially abundant between CRC patients and healthy individuals.

## Data Availability

The original contributions presented in this study are included in the article. Further inquiries can be directed to the corresponding author.

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
