# Peer review of "Metagenomic Signatures of Colorectal Cancer in the Jordanian Population: A Regional Case-Control Analysis Using 16S rRNA Profiling"

_microorganisms, 2025, doi:10.3390/microorganisms13081963_

Round 1

Reviewer 1 Report

Comments and Suggestions for Authors
  1. The control of healthy individuals is sourced from GutfeedingKB, rather than the healthy population in Jordan, which may lead to biased results due to factors such as different regions, diets, and genetic differences. Suggest adding local health controls in Jordan, or at least discussing the limitations of using this database as a control and its impact on the research results.
  2. It is suggested that the original sequencing data (such as NCBI login number) should be made public in the manuscript.
  3. If clinical data such as CRC patient staging and treatment plans can be supplemented, and the correlation analysis between clinical characteristics and microbiota can be analyzed, the results will be more comprehensive.
  4. The resolution of the 16S V3-V4 region is insufficient, such as the inability to distinguish E. coli pks and pathogenic strains, and the limitations of the technology need to be explained in the discussion.
  5. Lines 134-135 are not a complete sentence and need to be checked and modified.
  6. There is a formatting error in lines 243-248.
  7. The P-value should be italicized.
  8. Without Figure 5? Is Figure 6 not cited in the text?
  9. The writing style of GutFeelingKB should be consistent throughout the manuscript.
  10. Please carefully review reference number 32 and suggest replacement.

Author Response

Comment 1:
The control of healthy individuals is sourced from GutFeelingKB, rather than the healthy population in Jordan, which may lead to biased results due to factors such as different regions, diets, and genetic differences. Suggest adding local health controls in Jordan, or at least discussing the limitations of using this database as a control and its impact on the research results.

Response:
We thank the reviewer for highlighting this important limitation. We agree that the use of GutFeelingKB as a control group introduces potential bias related to demographic, geographic, and dietary differences. Importantly, GutFeelingKB was built using IRB-approved samples from well-characterized healthy individuals. Participants were carefully selected based on strict inclusion/exclusion criteria (e.g., age, health status, non-smoking, no recent antibiotic use, no recent illness or international travel), ensuring a homogenous and stable reference group. The study also included detailed dietary monitoring through 7-day food journals analyzed with the Nutrition Data System for Research (NDSR). Achieving this level of dietary and participant control would not have been practical for a healthy control cohort in Jordan under current resource and logistical constraints.

We acknowledge that species-level microbiome differences between populations may exist. However, as noted by the GutFeelingKB team, studies comparing healthy microbiota across different populations have reported reasonable consistency at higher taxonomic levels, particularly at the genus and family levels. Therefore, we believe GutFeelingKB remains a scientifically valid and appropriate reference for comparative analysis in our study. We have expanded the discussion of this limitation in the manuscript. We specifically address how these differences may influence microbial composition and interpretation of our findings. In future work, we will prioritize recruitment of local controls to strengthen comparative analysis.

We expanded the Discussion to include the following:

“In this study, we utilized the GutFeelingKB database as a source of microbiome data from healthy control individuals[53]. While GutFeelingKB provides a comprehensive catalog of gut microbiota profiles from healthy populations, its use as a control group introduces interpretive challenges, particularly because regional, dietary, and demographic variations can significantly influence the gut microbiome independent of disease status. In the absence of local healthy controls, it becomes challenging to ascertain whether the observed microbial alterations stem from colorectal cancer (CRC) pathology or pre-existing differences between the Jordanian and Western populations represented within GutFeelingKB. Furthermore, the lack of exhaustive metadata from the GutFeelingKB cohort restricts the capacity to conduct sensitivity analyses or appropriately match controls. Such limitations necessitate caution in attributing observed differences solely to CRC. Notwithstanding these constraints, employing GutFeelingKB represents a pragmatic approach given the unavailability of locally collected Jordanian healthy samples. The pronounced distinctions observed between CRC and GutFeelingKB profiles, such as significant disparities in Bacteroides abundance, likely surpass what technical variation could account for, indicating genuine disease-related shifts. In light of these factors, it is essential for future research endeavors to prioritize the recruitment of healthy Jordanian participants, ideally matched for parameters such as age, gender, and dietary habits, to serve as controls. This approach will enhance the validity of microbiome comparisons and facilitate the distinction between alterations that are truly indicative of disease and those that can be attributed to regional baseline differences.”

Comment 2:
It is suggested that the original sequencing data (such as NCBI login number) should be made public in the manuscript.

Response:
We appreciate this suggestion and fully support open data practices. We have created a BioProject on NCBI (ID: PRJNA1289688) and submitted the FASTQ files to the SRA. We are now waiting for the SRA accession numbers. The accession number will be added to the revised manuscript as soon as it becomes available. Moreover, we have uploaded the raw data to the cloud drive and you can find it on the following link:  https://drive.google.com/drive/folders/1lL38quUdq1OtHw6DLxW07PBLG_mqXmtG

Comment 3:
If clinical data such as CRC patient staging and treatment plans can be supplemented, and the correlation analysis between clinical characteristics and microbiota can be analyzed, the results will be more comprehensive.

Response:
We acknowledge the value of integrating clinical metadata to enhance the depth of analysis. However, due to ethical and anonymization protocols in this study, individual clinical data such as cancer staging and treatment history were not collected or linked to the microbiome samples. We have clarified this limitation in the Methods and Discussion sections and recommend that future studies incorporate detailed clinical metadata to enable such correlations.

We added the following text to the Methods part:

“Due to anonymization protocols, individual clinical data (including cancer stage and treatment history) were not available for correlation analyses with microbiome profiles. Future investigations should strive to collect and integrate clinical metadata to better elucidate relationships between microbial alterations and patient characteristics.”

Comment 4:
The resolution of the 16S V3-V4 region is insufficient, such as the inability to distinguish E. coli pks and pathogenic strains, and the limitations of the technology need to be explained in the discussion.

Response:
Thank you for raising this technical consideration. We have elaborated in the Discussion on the inherent limitations of 16S rRNA V3–V4 amplicon sequencing, particularly its inability to distinguish pathogenic from commensal strains (such as E. coli with/without the pks island) and the reduced capacity for strain-level resolution.

We added the following text to the Discussion:

“A key limitation of 16S rRNA V3–V4 amplicon sequencing is its insufficient resolution for species- and strain-level identification. This precludes differentiation between pathogenic and non-pathogenic strains, such as Escherichia coli carrying the pks island, which are relevant to CRC pathogenesis. Future studies employing shotgun metagenomics or targeted PCR/qPCR assays could provide more granular taxonomic and functional insights.”

Comment 5:
Lines 134-135 are not a complete sentence and need to be checked and modified.

Response:
We appreciate this careful review. We have revised lines 134–135 to ensure the text is complete and grammatically correct.

Comment 6:
There is a formatting error in lines 243-248.

Response:
Thank you for noting this. We have reviewed and corrected the formatting error in lines 243–248.

Comment 7:
The P-value should be italicized.

Response:
We have revised the manuscript so that all instances of “P” (when reporting p-values) are now italicized, as per journal style.

Comment 8:
Without Figure 5? Is Figure 6 not cited in the text?

Response:
Thank you for identifying this discrepancy. We have checked the figure sequence and ensured that all figures are present, properly numbered, and cited in the main text accordingly.

Comment 9:
The writing style of GutFeelingKB should be consistent throughout the manuscript.

Response:
We have reviewed all mentions of “GutFeelingKB” and ensured consistency in capitalization, formatting, and abbreviation throughout the manuscript.

Comment 10:
Please carefully review reference number 32 and suggest replacement.

Response:
Upon reviewing reference 32, we note it is a retracted article (from Wiley). We agree this should be replaced with a more reliable and current reference.

Replacement:

Li, G., Zhao, D., Ouyang, B., Chen, Y., & Zhao, Y. (2025). Intestinal microbiota as biomarkers for different colorectal lesions based on colorectal cancer screening participants in the community. Frontiers in Microbiology, 16, 1529858.

Reviewer 2 Report

Comments and Suggestions for Authors

This manuscript presents an investigation into the gut microbiota composition in Jordanian patients with colorectal cancer (CRC). It adds an important dataset to the growing body of microbiome-oncology literature, particularly from Middle Eastern populations, which are currently underrepresented in this field. However, several significant issues should be addressed to strengthen the manuscript before it can be considered for publication.

Major comments

The GutFeelingKB dataset was used as a control group. There is no doubt that this is a pragmatic solution. However, it introduces substantial confusion due to geographic, dietary, and demographic differences. The authors should discuss in greater detail how this limitation affects the interpretation of results. Also, they should consider performing sensitivity analyses if any metadata from GutFeelingKB is available.

The manuscript indicates that samples were collected from already diagnosed and treated CRC patients. Treatment (e.g., chemotherapy, antibiotics), as well as dietary changes, can significantly alter microbiota composition. Please clarify the clinical status and treatment history of patients at the time of sampling and discuss how these factors may confound microbiota differences attributed to CRC itself.

Fecal samples provide a general overview of gut microbiota but may miss tumor-adherent microbes, such as Fusobacterium nucleatum, which are often enriched in mucosal biopsies. This limitation should be acknowledged and discussed: How does the sampling approach influence findings, especially the absence of taxa commonly reported in other CRC studies?

It is unclear whether multiple testing corrections were applied to the Wilcoxon and LDA analyses. Please clarify and, if not already done, apply appropriate corrections (e.g., FDR or Bonferroni) to minimize Type I errors.

The manuscript discusses microbial metabolites and mechanisms (e.g., SCFAs, genotoxins) based solely on taxonomic shifts. Consider employing predictive metagenomic tools (e.g., PICRUSt2) or temper conclusions by clearly stating that functional interpretations are speculative in the absence of direct functional data.

The authors note the absence of Fusobacterium nucleatum in CRC samples and suggest this might reflect regional microbiome differences in the Jordanian population. However, the control data used (GutFeelingKB) originates from Western populations. Without parallel analysis of healthy Jordanian individuals, it is not possible to determine whether F. nucleatum is absent due to regional baseline characteristics or whether its absence is an anomaly in CRC pathology. Please write this interpretation more cautiously; the limitation of using non-local controls should be explicitly stated. Future work needs to prioritize the recruitment of healthy Jordanian controls.

Minor Comments

Figure 1. Is the average of all samples or a representative sample?

Figure 4. The color codes of CRC and healthy bacterial genera do not match, which adds to the confusion when readers want to compare

Please ensure that all references are updated and formatted following the journal's style.

Comments on the Quality of English Language

The quality of the English is generally understandable; however, there are several grammatical errors, strange phrasings, and repetitive sentences that make it difficult to read. While scientific terms are used accurately, the phrasing can sometimes be too wordy or redundant, particularly in the results and discussion sections.

I recommend seeking professional language editing to enhance sentence structure and consistency and improve the overall academic tone of the manuscript. It is also important to carefully revise the figure legends to ensure correct grammar and clarity, making them self-explanatory without relying on the main text.

Author Response

Comment 1:
The GutFeelingKB dataset was used as a control group. There is no doubt that this is a pragmatic solution. However, it introduces substantial confusion due to geographic, dietary, and demographic differences. The authors should discuss in greater detail how this limitation affects the interpretation of results. Also, they should consider performing sensitivity analyses if any metadata from GutFeelingKB is available.

Response:
We thank the reviewer for highlighting this crucial limitation. We have expanded the Discussion to explicitly address how the use of GutFeelingKB as a control group may affect interpretation, emphasizing potential confounding due to regional, dietary, and demographic mismatches. Unfortunately, detailed metadata from GutFeelingKB (e.g., for stratified or sensitivity analyses) is not publicly available for all reference samples. We have now stated this in the revised manuscript and clarified its impact on our analyses. Future work will prioritize recruitment of local controls for more precise comparisons.

We added to the Discussion:

“In this study, we utilized the GutFeelingKB database as a source of microbiome data from healthy control individuals[53]. While GutFeelingKB provides a comprehensive catalog of gut microbiota profiles from healthy populations, its use as a control group introduces interpretive challenges, particularly because regional, dietary, and demographic variations can significantly influence the gut microbiome independent of disease status. In the absence of local healthy controls, it becomes challenging to ascertain whether the observed microbial alterations stem from colorectal cancer (CRC) pathology or pre-existing differences between the Jordanian and Western populations represented within GutFeelingKB.

Furthermore, the lack of exhaustive metadata from the GutFeelingKB cohort restricts the capacity to conduct sensitivity analyses or appropriately match controls. Such limitations necessitate caution in attributing observed differences solely to CRC. Notwithstanding these constraints, employing GutFeelingKB represents a pragmatic approach given the unavailability of locally collected Jordanian healthy samples. The pronounced distinctions observed between CRC and GutFeelingKB profiles, such as significant disparities in Bacteroides abundance, likely surpass what technical variation could account for, indicating genuine disease-related shifts. In light of these factors, it is essential for future research endeavors to prioritize the recruitment of healthy Jordanian participants, ideally matched for parameters such as age, gender, and dietary habits, to serve as controls. This approach will enhance the validity of microbiome comparisons and facilitate the distinction between alterations that are truly indicative of disease and those that can be attributed to regional baseline differences..”

Comment 2:
The manuscript indicates that samples were collected from already diagnosed and treated CRC patients. Treatment (e.g., chemotherapy, antibiotics), as well as dietary changes, can significantly alter microbiota composition. Please clarify the clinical status and treatment history of patients at the time of sampling and discuss how these factors may confound microbiota differences attributed to CRC itself.

Response:
We appreciate this important point. Due to anonymization protocols and the retrospective nature of sample collection, individual clinical status and treatment histories (such as chemotherapy or antibiotic use) were not available. We have clarified this limitation in the Methods and Discussion sections and have explicitly discussed how post-diagnosis treatment and lifestyle changes may confound microbiome findings attributed to CRC alone.

We added to the Methods:

“Due to anonymization protocols, individual clinical data (including cancer stage and treatment history) were not available for correlation analyses with microbiome  profiles. Future investigations should strive to collect and integrate clinical metadata to better elucidate relationships between microbial alterations and patient characteristics.”

We added to the Discussion:

“It is important to note that fecal samples were obtained from patients who had already been diagnosed and potentially treated for CRC. Treatments such as chemotherapy, antibiotics, and dietary modifications can significantly alter the gut microbiome. Without detailed clinical metadata, it is not possible to distinguish microbiome changes caused by CRC itself from those induced by treatment or lifestyle adjustments. Future studies should collect and integrate these clinical details to disentangle disease-related microbiome shifts from treatment effects.”

Comment 3:
Fecal samples provide a general overview of gut microbiota but may miss tumor-adherent microbes, such as Fusobacterium nucleatum, which are often enriched in mucosal biopsies. This limitation should be acknowledged and discussed: How does the sampling approach influence findings, especially the absence of taxa commonly reported in other CRC studies?

Response:
We fully agree and have added discussion to acknowledge the limitations of using fecal samples versus mucosal biopsies, particularly regarding detection of tumor-adherent bacteria such as Fusobacterium nucleatum.

We added to the Discussion:

“Another limitation is that this study relied exclusively on fecal samples. While stool samples provide an accessible snapshot of the gut microbiome, they may underrepresent or miss tumor-adherent bacteria that preferentially colonize the mucosal surface, such as Fusobacterium nucleatum. The absence of such taxa in our results could be due, at least in part, to our sampling strategy. Future studies should incorporate both fecal and tissue-associated samples to obtain a more comprehensive view of CRC-associated microbiota.”

Comment 4:
It is unclear whether multiple testing corrections were applied to the Wilcoxon and LDA analyses. Please clarify and, if not already done, apply appropriate corrections (e.g., FDR or Bonferroni) to minimize Type I errors.

Response:

Thank you for this important statistical point. FDR correction was applied to the Wilcoxon test results using the Benjamini–Hochberg method. The results remained consistent, indicating that the significant genera are robust to correction for multiple testing.

Comment 5:
The manuscript discusses microbial metabolites and mechanisms (e.g., SCFAs, genotoxins) based solely on taxonomic shifts. Consider employing predictive metagenomic tools (e.g., PICRUSt2) or temper conclusions by clearly stating that functional interpretations are speculative in the absence of direct functional data.

Response:
We agree that functional inferences from 16S rRNA data are indirect and speculative. As predictive metagenomics (e.g., PICRUSt2) was not performed, we have revised the manuscript to temper conclusions about metabolic mechanisms, clearly stating that these are inferred based on taxonomic composition and should be validated in future studies using direct functional profiling.

We added to the Discussion:

“In this study, functional interpretations regarding microbial metabolites (e.g., SCFAs, genotoxins) are speculative and based solely on the known metabolic capabilities of identified taxa, not on direct functional measurements. Future research should incorporate predictive metagenomics (such as PICRUSt2) or shotgun metagenomic sequencing to more accurately characterize functional alterations in CRC-associated microbiota.”

Comment 6:
The authors note the absence of Fusobacterium nucleatum in CRC samples and suggest this might reflect regional microbiome differences in the Jordanian population. However, the control data used (GutFeelingKB) originates from Western populations. Without parallel analysis of healthy Jordanian individuals, it is not possible to determine whether F. nucleatum is absent due to regional baseline characteristics or whether its absence is an anomaly in CRC pathology. Please write this interpretation more cautiously; the limitation of using non-local controls should be explicitly stated. Future work needs to prioritize the recruitment of healthy Jordanian controls.

Response:

We appreciate this important clarification and have revised the Discussion to interpret the absence of Fusobacterium nucleatum with greater caution, explicitly noting the limitation imposed by the lack of local control data.

We added to the Discussion:

“The absence of Fusobacterium nucleatum in our CRC samples should be interpreted with caution, as the comparison group was sourced from GutFeelingKB, a Western population. Without analysis of healthy Jordanian controls, it is unclear whether the absence reflects regional baseline differences, methodological limitations, or a true deviation from global CRC patterns. Future research must prioritize recruitment of local, matched healthy controls to resolve this uncertainty.”

Comment 7:
Figure 1. Is the average of all samples or a representative sample?

Response:
Thank you for your observation. Figure 1 represents the average relative abundance of microbial phyla across all samples within each group (CRC and healthy controls), not a single representative sample. We have clarified this in the revised figure legend and main text.

Revised legend:

“Figure 1. Average gut microbiota composition at the phylum level across all CRC (n=42) and healthy control samples. Values represent the mean relative abundance for each group.”

Comment 8:
Figure 4. The color codes of CRC and healthy bacterial genera do not match, which adds to the confusion when readers want to compare

Response:
We appreciate your attention to detail. We have revised Figure 4 to ensure consistent color coding for bacterial genera between CRC and healthy groups, improving clarity and comparability for readers.

Comment 9:
Please ensure that all references are updated and formatted following the journal's style.

Response:
Thank you for highlighting this point. We have carefully reviewed and updated all references to ensure they are current and properly formatted in accordance with the journal’s requirements.

Comment 10:
The quality of the English is generally understandable; however, there are several grammatical errors, strange phrasings, and repetitive sentences that make it difficult to read. While scientific terms are used accurately, the phrasing can sometimes be too wordy or redundant, particularly in the results and discussion sections.

Response:
We thank the reviewer for this valuable feedback. The manuscript has been thoroughly revised for grammar, clarity, and conciseness. We have removed repetitive or awkward phrasing, improved sentence structure, and enhanced the overall academic tone, with special attention to the Results and Discussion sections.

Comment 11:
I recommend seeking professional language editing to enhance sentence structure and consistency and improve the overall academic tone of the manuscript. It is also important to carefully revise the figure legends to ensure correct grammar and clarity, making them self-explanatory without relying on the main text.

Response:
We appreciate this recommendation. We have conducted a careful, line-by-line revision to improve the English language and academic tone. All figure legends have been rewritten to be clear, grammatically correct, and self-explanatory, providing essential details without reliance on the main text.

Reviewer 3 Report

Comments and Suggestions for Authors

The paper "Metagenomic Signatures of Colorectal Cancer in the Jordanian Population" is interesting and well structured, but needs some clarifications:

1. Non-local controls (GutFeelingKB):

The control group is composed of data from the GutFeelingKB database, which collects healthy non-Jordanian subjects. This introduces potential geographic, cultural, dietary and genetic biases.

The authors themselves admit this, but the impact could be significant enough to compromise comparative validity

2. Absence of clinical data in CRC patients:

No age, sex, tumor stage, diet or current treatments of CRC patients are reported, which could strongly influence the microbiome.

Anonymization is important, but some aggregate data would have been useful.

3. No control in the statistics for diet, BMI, antibiotic or probiotic use, all known influencers of the gut microbiome.

4. The absence of a known CRC biomarker (F. nucleatum) is interesting, but not thoroughly investigated

Author Response

Comment 1:

The control group is composed of data from the GutFeelingKB database, which collects healthy non-Jordanian subjects. This introduces potential geographic, cultural, dietary and genetic biases. The authors themselves admit this, but the impact could be significant enough to compromise comparative validity.

Response:
We appreciate the reviewer’s careful consideration of this limitation. We have further expanded the Methods and Discussion sections to clearly state the risk of confounding due to population, dietary, and genetic differences between the Jordanian CRC group and the non-local GutFeelingKB controls. We explicitly acknowledge that these differences could influence observed microbial patterns and caution that direct causality should not be inferred. We have also reinforced in the manuscript’s conclusion and limitations that future research must include locally recruited, demographically matched controls to ensure the validity of comparative analyses.

Comment 2:

No age, sex, tumor stage, diet or current treatments of CRC patients are reported, which could strongly influence the microbiome. Anonymization is important, but some aggregate data would have been useful.

Response:
We thank the reviewer for this important point. Due to the study’s ethical protocols and complete anonymization, we were unable to collect or report aggregate clinical data (age, sex, tumor stage, diet, treatment history) for CRC patients. We have clarified this limitation in both the Methods and Discussion sections and explicitly state that this precluded stratified or subgroup analyses. In future work, we will seek to obtain and report aggregate clinical metadata while maintaining patient confidentiality.

Comment 3: No control in the statistics for diet, BMI, antibiotic or probiotic use, all known influencers of the gut microbiome.

Response:
We fully agree that factors such as diet, BMI, antibiotic, and probiotic use are major determinants of gut microbiome composition. Unfortunately, due to anonymization and lack of metadata, it was not possible to control for these variables in our statistical analyses. We have further emphasized this limitation in the revised Discussion, and will prioritize collection of such information in future studies.

Comment 4: The absence of a known CRC biomarker (F. nucleatum) is interesting, but not thoroughly investigated

Response:
We thank the reviewer for highlighting this point. We have expanded the Discussion to further analyze the possible explanations for the absence of Fusobacterium nucleatum in our CRC cohort. We now include a more detailed discussion of methodological, regional, and technical factors that may account for this observation, and we stress the need for future studies to confirm this result using both fecal and mucosal samples, as well as complementary molecular approaches (e.g., qPCR, metagenomics).

Reviewer 4 Report

Comments and Suggestions for Authors

This is a well-executed study. The article is scientifically sound,making a valuable contribution to the field of colorectal cancer and microbiome research.

Minor concerns 

The authors should clarify why the Jordanian population was specifically chosen for this study. While the novelty of studying an underrepresented region is implied, the rationale would be stronger if explicitly stated in the introduction. A brief paragraph discussing factors such as distinct dietary habits, environmental exposures, genetic background, or regional CRC trends would provide important context and justify the population choice.

The manuscript does not include a locally recruited control group. Instead, it relies on the GutFeelingKB database, which consists of non-local, unmatched healthy individuals. This is a significant limitation that should be more explicitly acknowledged and discussed in both the Methods and Discussion sections, particularly in relation to potential biases due to population, lifestyle, and technical differences.

The manuscript lacks important patient metadata, such as age, sex, cancer stage, and comorbidities, which limits the ability to perform stratified or subgroup analyses. This limitation should be clearly acknowledged and discussed in the Limitations section of the manuscript.

Author Response

Comment 1:
The authors should clarify why the Jordanian population was specifically chosen for this study. While the novelty of studying an underrepresented region is implied, the rationale would be stronger if explicitly stated in the introduction. A brief paragraph discussing factors such as distinct dietary habits, environmental exposures, genetic background, or regional CRC trends would provide important context and justify the population choice.

Response:
Thank you for this important suggestion. We agree that an explicit rationale for selecting the Jordanian population will add value and context. We have added a dedicated paragraph to the Introduction, highlighting the region’s unique dietary, genetic, and environmental factors, as well as the lack of local microbiome data in CRC, to clearly justify the study population.

We added to the Introduction:

“Jordan represents a unique and underrepresented population for CRC microbiome research due to its distinct dietary patterns, environmental exposures, and genetic background. Traditional Jordanian diets, characterized by high consumption of olive oil, whole grains, legumes, and fermented dairy products, differ markedly from Western dietary profiles and may shape a unique gut microbial landscape. Additionally, regional variations in antibiotic use, water sources, and environmental exposures may further impact gut microbiota composition. The Arab population, including Jordanians, also exhibits specific genetic polymorphisms that may influence host-microbiota interactions and CRC susceptibility. Despite rising CRC incidence in the region, there is a significant gap in microbiome data from Middle Eastern populations. By studying the Jordanian cohort, this research aims to address this gap and contribute region-specific insights to the global understanding of CRC-associated microbiota.”

Comment 2:
The manuscript does not include a locally recruited control group. Instead, it relies on the GutFeelingKB database, which consists of non-local, unmatched healthy individuals. This is a significant limitation that should be more explicitly acknowledged and discussed in both the Methods and Discussion sections, particularly in relation to potential biases due to population, lifestyle, and technical differences.

Response:
We agree with the reviewer. We have expanded the Methods and Discussion sections to explicitly acknowledge this limitation, detailing the potential biases introduced by using a non-local control group and clarifying how this may affect the interpretation of our findings.

Comment 3:
The manuscript lacks important patient metadata, such as age, sex, cancer stage, and comorbidities, which limits the ability to perform stratified or subgroup analyses. This limitation should be clearly acknowledged and discussed in the Limitations section of the manuscript.

Response:
Thank you for this observation. We have now clearly stated in the manuscript’s Limitations section that anonymization protocols prevented the collection of individual patient metadata, and this precluded stratified or subgroup analyses.

Round 2

Reviewer 2 Report

Comments and Suggestions for Authors

The revised manuscript shows substantial improvement in clarity, transparency, and scientific rigor. The authors have addressed all major and minor concerns appropriately. Below are my final observations and minor suggestions for further strengthening the manuscript:

  • The limitations of using GutFeelingKB as a control group are now clearly acknowledged, and the discussion is well-balanced regarding confounding effects due to demographic mismatch.
  • The authors have carefully qualified their interpretations of Fusobacterium nucleatum absence, avoiding over-speculation and calling for future work with local controls.
  • Limitations originating from post-diagnosis sampling and lack of treatment metadata are now transparently discussed in both the Methods and Discussion sections.
  • Confirmation of FDR correction strengthens the robustness of the statistical analysis.
  • The English language has notably improved, and figure legends are now self-contained and informative.

Minor suggestions

  1. Although the GutFeelingKB metadata is noted as unavailable, even a brief mention of what metadata is available (e.g., age range, sex) may help contextualize the comparability, if possible.
  2. Since the metabolic role of SCFA-producing taxa is discussed, a brief mention of how SCFA concentrations could be validated in future studies would be helpful to link taxonomic and functional data.
  3. A final proofread may help catch minor typographic inconsistencies (e.g., spacing, punctuation).